# DEM Generation from Fixed-Wing UAV Imaging and LiDAR-Derived Ground Control Points for Flood Estimations

**DOI:** 10.3390/s19143205

**Published:** 2019-07-20

**Authors:** Jairo R. Escobar Villanueva, Luis Iglesias Martínez, Jhonny I. Pérez Montiel

**Affiliations:** 1Grupo de Investigación GISA, Facultad de Ingeniería, Universidad de La Guajira, Km 5 Vía a Maicao, Riohacha 440007, Colombia; 2Escuela Técnica Superior de Ingenieros de Minas y Energía, Universidad Politécnica de Madrid. Departamento de Explotación de Recursos Minerales y Obras Subterráneas. C/ Ríos Rosas, 21, 28003 Madrid, Spain

**Keywords:** UAV, fixed-wing, LiDAR, ground control point, DEM, accuracy, floods

## Abstract

Geospatial products, such as digital elevation models (DEMs), are important topographic tools for tackling local flood studies. This study investigates the contribution of LiDAR elevation data in DEM generation based on fixed-wing unmanned aerial vehicle (UAV) imaging for flood applications. More specifically, it assesses the accuracy of UAV-derived DEMs using the proposed LiDAR-derived control point (LCP) method in a Structure-from-Motion photogrammetry processing. Also, the flood estimates (volume and area) of the UAV terrain products are compared with a LiDAR-based reference. The applied LCP-georeferencing method achieves an accuracy comparable with other studies. In addition, it has the advantage of using semi-automatic terrain data classification and is readily applicable in flood studies. Lastly, it proves the complementarity between LiDAR and UAV photogrammetry at the local level.

## 1. Introduction

Projected increases of heavy rainfall, based on climate models, are expected to aggravate local floods [1]. Thus, effective spatial tools are required by governments and societies to take action against increasing exposure to natural hazards [2]. Geospatial products, such as digital elevation models (DEMs) are useful topographic representations of space and have some specifics for flood studies [3,4]. In a further explanation, the DEM concept has been considered in the same way proposed by Polat et al. [5], which refers to DEM as the *Z*-dimension of the terrain digitally. There is also the digital surface model (DSM) that include natural and man-made objects. Highly detailed terrain modelling is usually produced from data obtained by active sensors such as airborne light detection and ranging (LiDAR) [6,7]. The bare ground representation in the form of DEM from these sources is the basis of urban [8,9,10] and peri-urban local flood studies [11]. LiDAR technology have as main advantages of its laser energy penetration to the ground, for instance, through canopies [7]; however, the cost and complexity of the data acquisition involved implies that such airborne data is not always easy to update, or sometimes is only partially available [5].

The development of photogrammetry techniques based on Structure from Motion Multi-View Stereo (SfM-MVS) of images acquired by low-cost cameras in unmanned aerial vehicle systems (micro UAV, ≤ 2 kg) has seen a strong development in the last decade [12]. This, together with the SfM-MVS processing in a single workflow, allows the DSM and DEM generation [13,14]. Although one of its main technical drawbacks is the time required for image processing [15]. However, the (relative) flexibility in image acquisition and the increasing offer of robust SfM-MVS processing software, have made UAV photogrammetry a valid low-cost alternative to piloted airborne LiDAR technology [5,14]. Studies show that image based UAV-derived DEMs are comparable to LiDAR for fluvial flood assessment applications [16,17,18,19], such as flood extent and volume estimations. Leitao et al. [20] showed that is possible to obtain detailed DEMs in urban environments from image based UAV platforms with quality comparable to LIDAR data (in terms of the difference between DEMs), and found that a realistic representation (resolution < 1 m), plays a fundamental role in the surface flow modeling. Therefore, it concludes that micro UAVs are a useful solution for describing urban landscapes. In the literature there are more examples of the functionality of UAV images in 2D urban hydrodynamic modelling [21,22], flood risk management and emergency response [23,24], and mapping of difficult-to-access areas [25]. It is widely accepted that the UAV-derived DEM accuracy from SfM-MVS, i.e., aerial or terrestrial photogrammetry processing, is influenced by flight design and planning factors, such as GSD (ground sample distance), inclusion (or not) of oblique images, sensor and camera lens, flight pattern and georeferencing method, etc. [26]. As a rule of thumb in UAV photogrammetry, vertical accuracy for a DEM obtained must be between one and three times the GSD of input imagery [27,28,29]. The impact of the georeferencing method on the accuracy of SfM-MVS products is critical, and also well established in the literature. Georeferencing is usually classified as: (i) direct, by means of UAV navigation (GPS/ IMU) instruments, and sometimes corrected in real time by GPS-RTK [30,31]; and (ii) indirect, through established ground control points [32]. Usually, the classical indirect georeferencing is considered the most accurate method [33]. Depending on the size of the UAV SfM-MVS project, ground control point determination can become a challenging task due to its time-intensive nature [34], and constraints found on the terrain [14]. 

Using existing elevation data, for example from airborne LiDAR, can be an alternative for georeferencing a UAV photogrammetric project. The literature shows the complementarity between LiDAR, as an alternative source for ground control points, and photogrammetry of airborne imagery [35,36,37,38]. Liu et al. [35] and James et al. [37] suggested the use of non-physical or virtual control points called "LiDAR-derived control points". This complementarity has been recently exploited with high-resolution imagery by a multi-rotor UAV platforms and terrestrial LiDAR data for 3D city modelling [39]. Persad et al. have proposed the use of LiDAR data and SfM-MVS image processing for modelling applications and DEM generation in a deltaic area [40,41]. However, there is no reference that shows the contribution of LiDAR data in DEM generation from fixed-wing UAV imagery, especially in flood assessment applications for estimation of areas and volumes. To validate the present contribution, it is necessary to compare UAV photogrammetric products with independent external elevation data [42,43] and with standard reference surfaces (e.g., LiDAR) for flood analysis [16,20]. If this complementarity is confirmed, the use of existing airborne remote sensing data (e.g., LiDAR databases) will prove to be an alternative georeferencing method for UAV researchers and flood specialists in order to obtain useful DEMs for local-level studies.

This work aims to investigate if LiDAR elevation data can be used in DEM generation from fixed-wing UAV imagery for flood applications. More specifically, it aims to (i) assess the accuracy achieved in DEM from the SfM-MVS processing chain using LiDAR-derived control points (LCPs); (ii) test the performance of two software applications used for DEM processing and (iii); compare flood estimations of volume and area between DEMs based on UAV and LiDAR data (reference). 

This paper is organized as follows: Section 2 describes the equipment employed and the methods followed: UAV surveys, LCP collection, image processing for DSM and DEM generation (SfM software comparisons), flood applications and assessment methods. Section 3 discusses the accuracy of the method, and then, after its validity is confirmed, the flood results. Section 4 is devoted to the discussion, and Section 5 presents the conclusions. 

## 2. Equipment and Methods

### 2.1. Case Study Description 

The broader location of the study area is the Northwest part of the coastal city of Riohacha (Colombian Caribbean), shown in Figure 1. This zone is bounded to the North by the Caribbean Sea, to the East by the Ranchería river delta and to the South and West by the inner-city districts. The area under study is a peri-urban zone with the lowest elevations in m above sea level (a.s.l.) within the city bounds, highly exposed to fluvial and pluvial urban floods. The terrain is characterized by a flat relief with occasional undulations; terrain elevations range from 8 m in the South, down to values near 0 m a.s.l. in the North (the coastline). The choice of the study area is based on two criteria: first, local interest due to its high exposure to severe storm flooding, as occurred in the course of El Niño-La Niña (2010/2011) [44,45]; second, the availability of previous hydrological and LiDAR elevation data, outlined by the red and blue lines in Figure 1. Additionally, the Riohacha context was selected as an emblematic case of emerging cities with high population growth rates and vulnerability to natural hazards thus raising scientific interest as shown in Nardini and Miguez [44].

### 2.2. Methodology

The methodology consists of four stages (Figure 2): Stage 1 describes the survey planning and LCPs determination; Stage 2 describes the image processing. Stages 3 and 4 are the main focus of this study. 

#### 2.2.1. Stage 1. Flight Planning, UAV-Based Surveys and LCPs (Proposed Method)

UAV flights were carried out within the framework of a collaborative humanitarian mapping project, “Mapatón por La Guajira” in February 2016 [46]. The main UAV flight objectives were: (1) To generate an orthophoto for object digitalization under the OpenStreetMap standard; (2) to survey an area greater than the one previously available in elevation and hydrological datasets for UAV flood research (Figure 1). The eBee™ SenseFly UAV platform was used in this study (Appendix A, Table A1). The system is equipped with consumer-grade on-board Global Positioning System (GPS) and inertial measurement Unit (IMU) modules for autonomous navigation and further image geotagging. The payload was a non-metric RGB camera with a CMOS active pixel sensor technology.

In order to find a balance between flight objectives and available funding, 3 flights were agreed upon with the UAV pilot. The resulting GSD for each flight was of ~ 10.3 cm/pixel (∼325 m AGL). A careful local risk analysis was carried out based on the Colombian UAV regulation ([47], September 2015) to assure that the flight design was safe. The assessment determined that the risk was minimal if executed in the early hours of the day when there are no piloted operations, and wind velocities are low. After inspecting the flight area and selecting a safe take-off and landing site, a platform inspection was carried out to rule out in-flight loss of control. After this, the programmed flights were executed by the pilot in accordance with the local UAV regulations. The flight plan execution, in-flight monitoring and camera trigger were managed by the eMotion flight mission software (Lausanne, Switzerland), which is able to handle multiple flights, and further geotag images during post-flight processing [48]. Images and flight data were stored and saved for processing at Stage 2. Technical parameters for the UAV-Based surveys are in Table 1.

Figure 3 shows general methodology of the proposed georeferencing procedure. Control point determination in Stage 1 was in turn divided into two steps. 

First, five ground control points (GCPs, Figure 1, green triangles) were determined with a millimetric precision total station (Figure 3a), and were used exclusively for the geometric correction of the orthomosaic by using the SfM processing chain (Figure 2). The orthomosaic resolution was set to 1 × GSD (10.3 cm/pixel); a maximum of 20.6 cm in horizontal error is assumed (2 × GSD) [26]. The georeferenced orthomosaic was used as input to extract positional *X,Y* control point coordinates. 

Second, with the aim of densifying well-defined control points, manual identification of surface features in the generated orthomosaic was performed, with the support of a shaded relief map [37] (Figure 3b). Positional *X,Y* control point coordinates of surface features were determined using ArcMap, from ArcGIS^®^ Software (Redlands, CA, USA). These control points were all characterized by being located on plain terrain, easily recognizable, and were taken as stable over the period between the capture of the LiDAR (2008) and UAV datasets (2016). Altimetric Z values of final LCPs were extracted from the reference LiDAR DEM (2008) for each control point (*X,Y)* coordinate labeled. The ArcGIS^®^ “Extract values to points” tool (with the option “Interpolate values at the point locations” selected), was used to assign the *Z* value for each control point. The total number of LCPs with known *X,Y,Z* coordinates distributed in the study area was set to 13; this value is near the recommended range suggested in the literature [16,49]. Lastly, these 13 LCPs were used in Stage 2 to generate the final DEMs (and DSMs) from SfM-MVS UAV-based image processing.

The LiDAR dataset was acquired in 2008 by the Colombian maritime national authority, DIMAR, for a coastline survey. The sensor used was a Leica ALS540 mounted on a Cessna 402B piloted platform, flying at an altitude of approximately 900 m AGL [50]. The horizontal LiDAR nominal point spacing reported was between 1 and 1.3 m, with a final density of approximately 0.7 points/m^2^. A “Model key point” ground classification was performed with the proprietary LiDAR contractor software MARS^®^ [51], with a density of around 0.25 points/m^2^. Based on the above metadata, we assume that the LiDAR dataset falls within the vertical accuracy of ASPRS 20 cm class [42] and quality level QL3 (≤ 20 cm RMSE), according to USGS specifications [52]. The expected accuracy of modern LiDAR (e.g., vertical < 10 cm RMSE, see Zhang et al. [53]), should fall within the high accuracy classes by USGS and ASPRS standards. The description of the orthometric height correction of the 1 m TIN-based LiDAR DEM (and DSM) are found in Escobar et al. [54]. The LiDAR-based DEM (as well as the DSM), was also included in the assessment in order to verify its accuracy as control surface for LCP extraction.

#### 2.2.2. Stage 2. Photogrammetric Processing of UAV-Based Imaging for DEM Generation

The Structure from Motion (SfM) algorithms of Multiple-View Stereo (MVS) images were used to process UAV-based images using LCPs. SfM-MVS photogrammetry is an automated image-based procedure that simultaneously determines 3D coordinates (structure) by the motion of the camera and is used for UAV-image based photogrammetry [15,27,55]. In contrast to conventional photogrammetry, SfM techniques allow the determination of internal camera parameters, as well as its pose, by using direct georeferencing [56]. In the present study, the most common SfM-MVS software suites were used: Agisoft PhotoScan^®^ Professional v.1.1.2, and Pix4Dmapper^®^ Pro v.4.1.23 [15,56]. Both are commercial packages that use similar algorithms [57,58], however, the motivations in this paper were to estimate the vertical precision and explore the differences in the processing and interaction of the user with the software. This was one of the fundamental reasons why these SfM-MVS tools were chosen. The suites use modified known algorithms similar to the widely used SIFT for feature matching and key-point extraction [59]. Usually, the exterior orientation is possible by employing geotagged data to perform a bundle block adjustment and an iterative Newton’s computer vision method (i.e., Gauss-Markov [60]). The georeferenced sparse point cloud is obtained first, then re-optimized with external control points (absolute georeferencing), and later densified by custom pixel matching autocorrelation MVS algorithms [61]. Details of the hardware used for image processing are in Appendix A (Table A1). The purpose of using both SfM-MVS suites is to generate accurate UAV-derived DEMs (and DSMs) employing the same input dataset of geotagged imagery and control points (LCPs). For the above, two strategies were tested: (a) PhotoScan used as a semi-automated chain process; (b) Pix4D used as a fully-automated process. The general photogrammetric processing is shown in Figure 2 (Stage 2). During the sparse cloud generation, no manual edition of outliers and wrong located points was done. The set of 13 LCPs (Figure 3b) was tagged manually on the imagery dataset for the re-optimization of the initial photogrammetric procedure. Processing settings for both strategies are shown in Table 2. 

Details for each strategy for DSM and DEM generation are described as follows: 

*PhotoScan:* The workflow followed was Agisoft’s protocol for DEM generation [62]. Based on the estimated camera positions, PhotoScan calculates depth information for each camera to be combined into a single dense point cloud [57]. Dense cloud points corresponding to permanent water bodies were edited out. The ground filtering algorithm applied was the two-step approach based on the adaptive TIN algorithm described by Axelsson [63]. This algorithm divides data into square grids wherein a temporary TIN model is created first, and then densified with new points by calculating distance and angle parameters. The process is iterated, guided by user criteria, until all points of the terrain model are classified by adjusting the settings of the parameters shown under”Classifying dense cloud points” in Table 2. DSMs and DEMs were rasterized based on mesh data. The final raster DEM was created from dense classified “ground” points. 

*Pix4Dmapper:* The software’s graphic user interface allows to follow the workflow in three steps [58]. The initial step involves a fully automatic iterative proprietary algorithm for bundle block adjustment and sparse cloud point generation (Step 1). A custom MVS matching algorithm is applied for sparse cloud densification (Step 2), and then, the inverse distance weighting algorithm (IDW) interpolation is used for DSM generation (Step 3). The raster DEM was generated by selecting the “Point cloud classification” parameter (Table 2). Terrain extraction and DEM generation is based on fully automatic custom machine learning algorithms that classify the dense point cloud generated in typical semantic class labels, e.g., bare earth, buildings, vegetation and roads [64]. The user has no prior control in the training of the classification algorithms.

#### 2.2.3. Stage 3. UAV-Derived DEM Accuracy Assessment

To assess how well the UAV-derived DEMs obtained in Stage 2 represent the ground truth, a number of high precision observations *(n* = 104) were employed. This dataset (2018) was based on traditional surveying (millimeter range precision) made available from the city’s urban planning office (data from a replacement sewage pipe project [65]). The location of these checkpoints is distributed in the south and north zones of the study area, and is limited to those areas with less probability of variation (Figure 4), mainly road or street intersections (yellow lines). 

This elevation dataset is taken as independent ground truth data in order to perform statistics of the difference (*ΔZ*) from the observed value of the checkpoint (*Z*) and the ones from UAV-derived DSMs and DEMs (PhotoScan, Pix4D). The *Z* elevation from the checkpoint, as well as those from the UAV and LiDAR-based models were extracted for the same horizontal checkpoint location using the ArcGIS extraction toolset “Extract multi values to points”. To make the comparison of elevations in models possible, the MAGNA-SIRGAS (EPSG 3116) coordinate system reference was applied to all data sets. 

Two methods were tested to assess DSM and DEM accuracy. The first method is based on the root mean square error, RMSE, which is commonly used under the assumption that the set of *{ΔZ}* is normally distributed and is located over open areas not prone to outlier influence [42]. The second method is a measure of accuracy based on robust estimators, suggested in Höhle et al. [43,66], the Normalized Median Absolute Deviation (NMAD): (1)$$NMAD\;=1.4826\;\cdot \;median_{i}\left(\left|\Delta Z_{i}-\;m_{\Delta Z}\right|\right),$$
where *m_ΔZ_* is the median of the errors, and *ΔZ_i_* are the individual errors. NMAD is thus proportional to the median of the absolute differences between errors and the median error. It is a measure of accuracy that does not require a priori knowledge of the error distribution function [67] and especially useful in non-open and built-up terrain [43]. 

Basic statistical analysis for *ΔZ*, such as mean, standard deviation and median were also employed, together with histograms and box plots; normality tests for error distributions (*ΔZ* residuals) were performed (Shapiro–Wilk test). 

In order to compare our DEM performance with similar studies, the assessed vertical error-to-GSD ratio was determined. This ratio is somewhat of a rule of thumb in the UAV photogrammetry literature based on RMSE. Its values range from approximately 1 to 3 times the GSD for correctly reconstructed models [27,68]. Assuming that the LiDAR reference accuracy from where the LCPs were extracted is ≤ 20 cm (Section 2.2.1), the expected absolute accuracy (RMSE) for the obtained models are within the range of 20.6 – 30.9 cm (from 2 to 3 times the GSD). Likewise, expected accuracy ranges based on NMAD are given by Bühler [29] and Zazo [67]. Therefore, the expected ranges are between 11.5–23.7 cm (i.e., relative accuracy from 1.1 to 2.3 times the GSD).

#### 2.2.4. Stage 4. Flood Estimations from UAV-Derived DEMs

Finally, to test the performance of flood estimations of UAV-derived DEMs with respect to the LiDAR reference [16,20], two criteria were considered: first, a comparison based on the calculation of flood volume (*V*) and area (*A*) for a historical extreme event; second, a comparison based on the similarity of flood accumulation. 

First: The percentage differences (errors) in area and volume (*V_DIF_, A_DIF_*) of UAV-based DEMs were calculated with respect to the LiDAR-based DEM for a local extreme flood event, according to Equations (3) and (4) [11]: (2)$$V_{DIF}=\;\;\frac{\left|V_{PhotoScan,Pix4D}-\;V_{LiDAR}\right|}{V_{LiDAR}}\;100\%$$

(3)$$A_{DIF}=\;\;\frac{\left|A_{\;PhotoScan,Pix4D}-\;A_{\;LiDAR}\right|}{A_{LiDAR}}\;100\%$$

Second: Another criterion used to test the performance of flood estimation is based on the similarity of areas and volumes between the LiDAR reference and UAV-derived DEMs for each sub-basin. *A* and *V* were estimated for 10 depth-filling time interval (timesteps). These timesteps were defined as blocks of discrete filling simulations (at a fixed interval), where the flood elevation was increased in each DEM until it reached its maximum for the extreme flood event. The 10 time steps were compared to the LiDAR reference using the Bray-Curtis (Sørensen) index [69] for each DEM. These normalized values range from 0 to 1, where 0 represents exact agreement between two flood estimation datasets.

The spatial modelling tool used to estimate flood volume and area for a given UAV-derived DEM was the r.lake.xy hydrological module from GRASS GIS [70]. This module fills the DEM with a water elevation (*H*) at a seed point. The seed point elevation (*Z_sp_*) for a given flood elevation *H* (*Z_sp_* + *h*) is approximately located at the lowest point for each DEM. The flood depth *(h)* was obtained from hydrodynamics simulations for a historical flood (which occurred on 18/09/2011) performed by a MODCEL© model [44,71]. Flood volumes and areas were estimated for each DEM sub-basin (Figure 1). The corresponding *H* inputs can be seen in Table 3.

## 3. Results

### 3.1. UAV-Derived DEM Accuracy Assessment

As can be seen in Figure 5, the errors obtained for the considered models (LiDAR, PhotoScan and Pix4D) are greater for DSMs than for DEMs, with the highest errors obtained with PhotoScan. This difference in dispersion values may be due to reprojection errors (1.97 in PhotoScan vs. 0.19 in Pix4D) and camera optimization (1.90% in Photoscan *vs.* 0.21% in Pix4D). 

The absolute value of the average error for the considered models is below 10 cm, except for the DSM obtained from PhotoScan, which is 16.5 cm (Figure 5 and Figure 6), although the sign of the average value indicates that the models are either above or below the employed checkpoints. In addition, normality tests indicate that errors do not follow a normal distribution (except PhotoScan DEM), which implies that the RMSE estimators are in this case not suitable. From this, it can be considered that the obtained mean value of the error is underestimated. It is kept here for comparison only. Given the lack of normality of the errors, robust estimators are applied (Table 4).

The median error for the different models considered is between 10 and 20 cm, being in all cases positive, which implies that the models are below the employed ground truth checkpoints. As for the mean error, the highest median is obtained for the PhotoScan DSM. For NMAD the obtained values are similar for all models, although slightly higher for the DSMs. Values obtained by robust estimators are adequate for built-up areas, and their use is preferred over standard estimators. 

Figure 7 compares absolute and relative values of the expected accuracy of each UAV model with the standard LiDAR reference. For DSMs the accuracy value of PhotoScan is lower than that of Pix4D but is found to be near the expected range. For DEMs the values are similar for all models and the accuracy of PhotoScan is remarkably improved. The average accuracy values lie within the expected range and are close to the observed ones in the LiDAR reference. DEM accuracy can be mostly explained by differences in the operator's interaction with the SfM-MVS software. 

Since LiDAR DEM vertical accuracy is 18 cm, it falls within ASPRS/USGS standards, and is also similar to an empirical assessment by Hodgson [72]. Furthermore, based on the accuracy of our results, we can conclude that the UAV photogrammetry georeferencing method applied is valid for DEM generation.

### 3.2. Flood Estimations from UAV-Derived DEMs

Table 5 shows the results of the flood estimates, measured by volume and area, for each of the sub-basins and DEM considered (LiDAR reference, PhotoScan and Pix4D SfM-MVS), and their corresponding area and volume error (Equations (2) and (3)). It is clear that, on average, flooded volumes and areas obtained for PhotoScan DEM are closer to those of LiDAR. Table 6 summarizes the estimates of similarity for the progression of the flood. The observed results confirm the above, since the progression of the flood for PhotoScan DEM is closer to the reference than Pix4D’s.

Figure 8 shows the flood maps for the different methods, for sub-basins 704 and 603. Figure 9 shows the corresponding time evolution of the flood. Flood maps for Sub-Basin 704 show that LiDAR shares features with both, Pix4D and PhotoScan: Whereas for Pix4D the flooding occurs mainly along the streets, for PhotoScan it forms a broad water surface.

Figure 9 shows that for the Pix4D model in Sub-Basin 704 the volumes obtained for each timestep are closer to those of the reference, although slightly higher. For PhotoScan, flooded volume estimations are always below the reference. In Sub-basin 603, where the pond is located, the flood extent of the Pix4D model is reduced to the water body, whereas PhotoScan’s is much closer to the reference. 

The main observed differences between DEMs are explained by the method of classifying and editing the dense point cloud. Models generated in a semi-automatic way (PhotoScan), where operator intervention is important, produce results much closer to the LiDAR reference than where the process is fully automatic (i.e., Pix4D). It can be seen that for Pix4D DEMs, part of the infrastructure (buildings, for example) remains in the final DEM.

## 4. Discussion

Our results show that airborne LiDAR-derived control points are useful in obtaining accurate DEMs from UAV-based RGB imaging, with a resolution of two times the pixel size of input imagery. PhotoScan offers better interactivity, especially in DEM generation. Although its DSM accuracy turned out slightly inferior than Pix4D's, it was compensated when DEM is generated. The UAV-based DEMs are in fact as accurate as LiDAR DEMs, and this is in agreement with the work of Polat and Uysal [5]. In general, DEMs obtained by a SfM-MVS processing chain are within the expected ranges reported in the literature (Figure 8). This is also confirmed when comparing the relative accuracies with the ones in Table A2 in the Appendix A. Therefore, the input of control points from airborne LiDAR to SfM-MVS processing of fixed-wing UAV imaging is justified [35,36,37]. Furthermore, our results contribute to broadening UAV photogrammetry applications when the determination of control points is a burden, for example, in emergency situations [23,24]. It also enables exploiting automatic integration, as shown in the literature [38,41,73,74]. In addition, it makes quick and efficient DEM generation possible, as well as to carry out multitemporal analysis, which is one of the main advantages of UAV platforms [75]. Finally, based on the trends of the abovementioned literature and results, the increasing offer of geospatial products is promising, especially in order to achieve UN Sustainable Development Goal 11, “Sustainable Cities and Communities” by 2030 [2]. 

Important limitations for the replication of the described method are the current international regulations for civil UAV operation, in particular, the flight altitude. However, by reducing it, similar or better accuracies (in relative terms) as those reported in the literature are to be expected at the expense of a smaller area coverage per flight (Appendix A, Table A2), and therefore, the need for longer flights [76]. This requires that the LiDAR reference and the pixel size of the UAV images maintain a relative accuracy of at least 2:1. For example, for a 5 cm pixel (~150 m AGL using the same equipment), the LiDAR must have a vertical accuracy equal to or less than 10 cm. The ever-increasing availability of terrestrial LiDAR elevation data can become an additional source of control points for SfM-MVS UAV photogrammetry, as has been recently shown in the literature [39,77]. 

Results for flood estimations compared to LiDAR show the usefulness of DEMs generated from SfM-MVS dense point clouds, when the user is actively involved in their classification. These findings are in agreement with those of Leitão et al. [20], Coveney et al. [16] and Schumann et al. [17], who based their comparisons on previous reference LiDAR surfaces. Outcomes of flood analysis showed the suitability of using DEMs from SfM-MVS as a tool to support local flood studies in urban catchments or peri-urban floodplains [75]. Specifically, this allows obtaining useful input elevation data for 2D hydrodynamic modelling in urban areas, as suggested by Yalcin et al. [21]. On the other hand, the estimation of extreme flood events makes it possible to investigate the generated DEM beyond the streets where the precise altimetric information was available. This warrants carrying out a more general UAV DEM assessment.

While discrepancies are evident between UAV models and the reference LiDAR, they are due to the DEM generation strategy, which is highly sensitive to the filtering method for ground point extraction and dense cloud edition. Flood map outputs show that for Pix4D DEM there exists a tendency of rendering a certain residual urban fabric, owing to deficient quality of the determination of the DEM by the software (Figure 8). The inclusion of residual urban fabric in the Pix4D-derived DEM influences the flood extent by volume displacement. Consequently, in Pix4D DEM, flooding tends to propagate along the streets, in contrast to PhotoScan DTM, where a broad water surface is observed. This agrees with the conclusions of Shaad et al. [78] and Hashemi-Beni et al. [23], who showed that fully automated ground extraction algorithms generate worse flood estimates than those obtained from a manual or semi-automated classification. A thorough user knowledge of the area, together with the availability of additional field data (profiles and/or observations of flood depths) is essential to ensure adequate utilization of ground filtering algorithms [79]. Discrepancies between flood assessments may also be due to the use of an outdated reference surface (e.g., LiDAR reference in 2008 vs. UAV surveys in 2016, as in our case), particularly in low-lying areas, where terrain variations or changes in hydraulic infrastructures (e.g., in channels or near box culverts) play an important role in flood propagation. The above suggests an opportunity to study terrain evolution with high-resolution UAV surveys [19]. 

This paper only solved the processing of UAV raw data, but differences in the resolution of UAV DEM and reference LiDAR could have an impact on flood estimates [16,20]. Further works might focus on finding an optimal DEM resolution by resampling methods for flood comparisons between UAV and LIDAR data. Additionally, the elevation dataset of the presented case study can be applied for the implementation of a local early warning system to estimate possible flood volume detection and water distribution in micro-morphology of streets. 

The main UAV advantage is their flexibility to acquire image data [20], especially for small to medium size areas (< 1 km^2^, up to 7 km^2^, see Table A2 in the Appendix A). On the other hand, the major disadvantages of UAV technology include a limited coverage area (e.g., flight time, payload and weather conditions) and the requirements for data processing. Piloted airborne platforms are more suited up to national scale surveys, while UAV is naturally better suited for an urban sub-basin scale. From the viewpoint of data processing, the larger size UAV project, the longer the time to processing. Processing time effort is about 45% of the UAV workflow [27], contrary to airborne LiDAR, in which 3D data is obtained automatically. 

Economic analyses by Jeunnette and Hart [80] concludes that piloted platforms lead to a lower cost of operation at 610 m AGL. The UAVs would become cost-competitive at approximately 305 m, flight height close to that used in the present study (~325 m AGL). Yurtseven [76] confirms the above at 350 m AGL, and also, it would be providing reasonable vertical accuracy and minimize the potential for systematic errors such as "doming effect" on the elevation products. 

The increasing operational capabilities of civil micro UAV will doubtlessly integrate with other technologies, as was here shown with LiDAR. However, current aviation safety regulations often pose limitations to research endeavors, especially because regulations seldom keep pace with technological development [47]. This means that UAV operators, society, authorities and industry have to continue working together towards a continuous improvement of local regulations [81,82]. It is expected that innovations in UAV safety will allow the next generation of fully integrated platforms to enter the airspace [83]. 

Finally, the use of LCPs as proposed in this work and according to James et al. [37], might involve important limitations, including: (i) availability of LiDAR data, some regions have either partial coverage or none at all, (ii) resolution of LIDAR data should be sufficiently detailed to allow the human operator to identify the superficial features, (iii) loss of information due to interpolation from raw point cloud data to grid data, and (iv) possible variations due to the temporal differences between LIDAR data acquisition and UAV surveys.

## 5. Conclusions

In this study the contribution of existing altimetric data of airborne LiDAR in DEM generation from UAV-based images (10.3 cm pixel size) for flood applications was investigated. Georeferencing based on LiDAR-derived control points has been applied, DEM accuracy assessed, and further applied to flood estimations. Floods from the corresponding UAV DEM were compared to those from a LiDAR DEM reference. 

The applied LCP georeferencing method contributes to obtaining DEMs with vertical accuracies comparable to those found in the literature, of approximately 2 times the pixel size of the input imagery. The DSM obtained with Pix4D is slightly more accurate than PhotoScan. However, the PhotoScan DEM is closer to the reference LiDAR, and therefore, more suitable for flood assessment applications (volume and area flooded estimations). In general, the feasibility of semi-automatically obtained UAV DEMs is confirmed. The hereby proved complementary nature between LiDAR and SfM-MVS photogrammetry will provide terrain modelers and flood scientists with an alternative tool for georeferencing their UAV (e.g., fixed-wing) photogrammetric products, in particular when ground control point determination is challenging. 

The expected applications of micro-UAV systems and the increasing supply of LiDAR datasets are promising for floods studies at the local level. Future work might focus on assessing the DEM accuracy of detailed terrestrial LiDAR-georeferenced UAV flights, and testing the impact of spatial resolution on flood estimates.

## Figures and Tables

**Figure 1 sensors-19-03205-f001:**
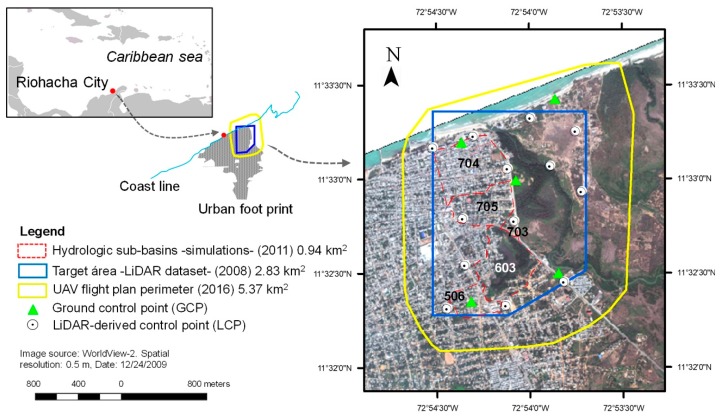
Map of the selected study area and location of control points for DEM generation. Hydrologic sub-basin (red dashed polygons) and numbers (bold type) were delineated and coded from local flood simulations performed by Nardini and Miguez [44].

**Figure 2 sensors-19-03205-f002:**
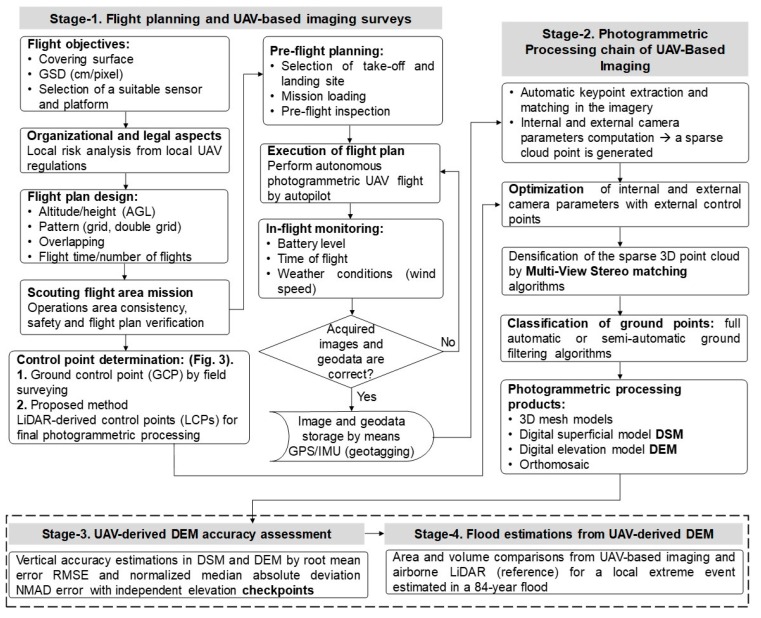
General outline of the methodology.

**Figure 3 sensors-19-03205-f003:**
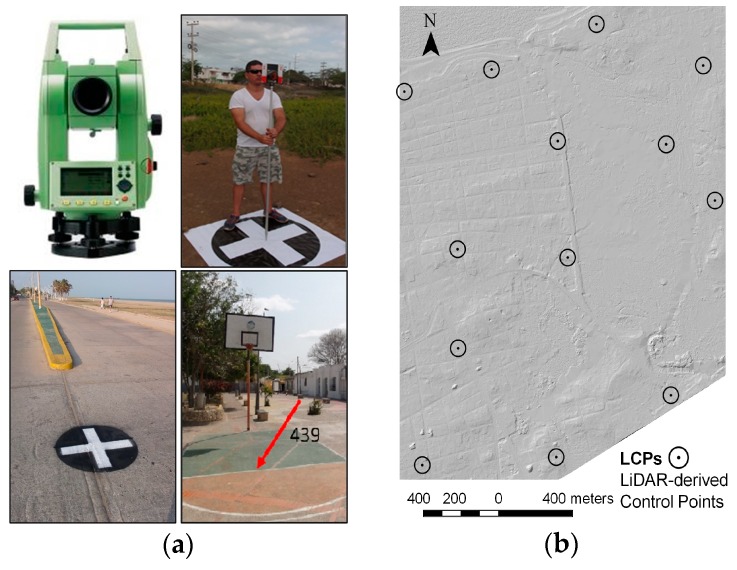
General methodology for ground control point (GCP) and LiDAR-derived control point (LCP) determination (method applied); (**a**) Leica TCR 403 total station used for GCP; (**b**) LiDAR DEM altimetric reference (displayed in shaded relief) with LCPs placed on study area.

**Figure 4 sensors-19-03205-f004:**
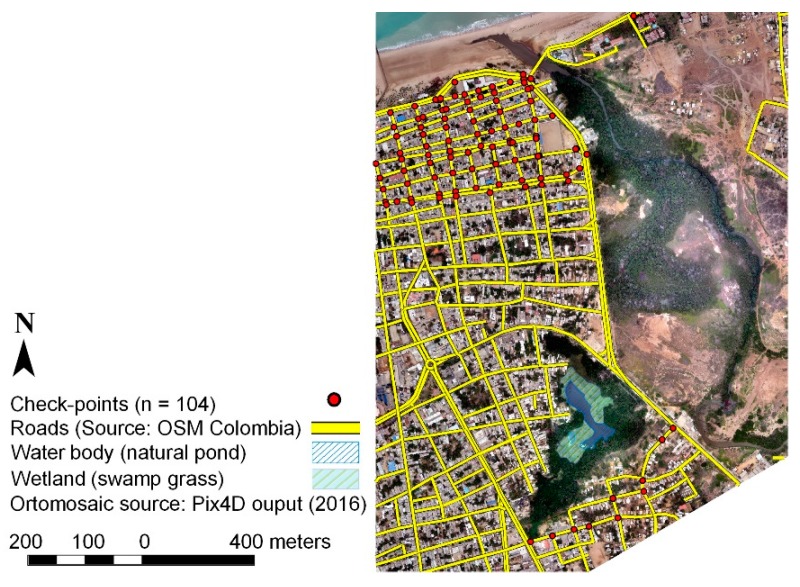
Checkpoint locations for DEM accuracy assessment.

**Figure 5 sensors-19-03205-f005:**
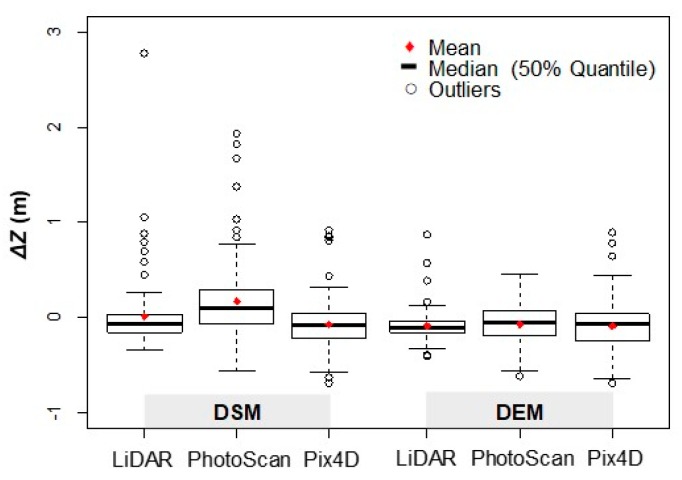
Box plots for comparison of *ΔZ* error for each model (No. of checkpoints = 104).

**Figure 6 sensors-19-03205-f006:**
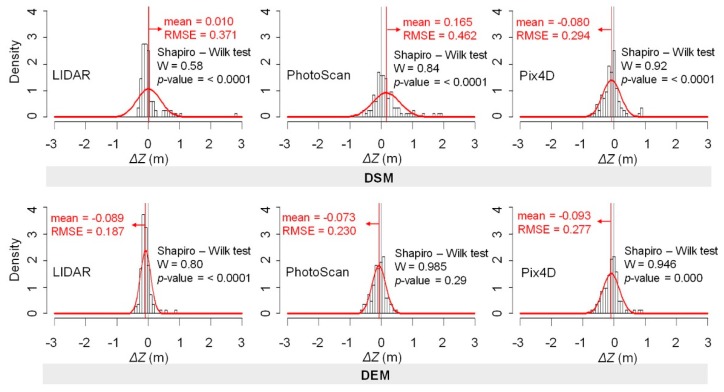
Histogram of the *ΔZ* values (n = 587) for each model. Superimposed on the histogram are the expected normal distribution curves with mean and RMSE estimated from all the data (red). The Shapiro-Wilk test results are shown (if *p*-value ≥ 0.05, *ΔZ* are normally distributed).

**Figure 7 sensors-19-03205-f007:**
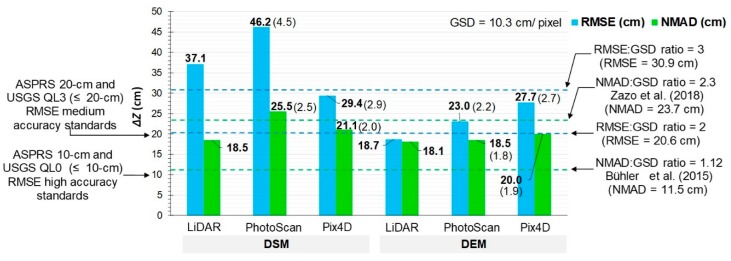
Accuracy (absolute and relative) of UAV-derived models, and comparison with LiDAR (no. of checkpoints = 104). Relative accuracy ratios are shown in brackets. USGS/ASPRS accuracy standards (**left**), as well as expected accuracy (**right**) are shown as horizontal dashed lines.

**Figure 8 sensors-19-03205-f008:**
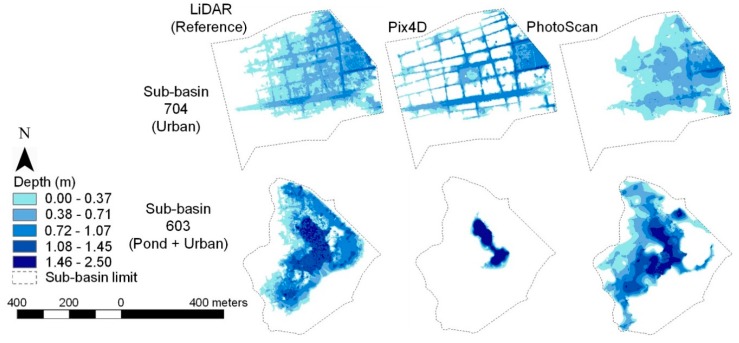
Flood extent to the corresponding DEM in 704 and 603 sub-basins.

**Figure 9 sensors-19-03205-f009:**
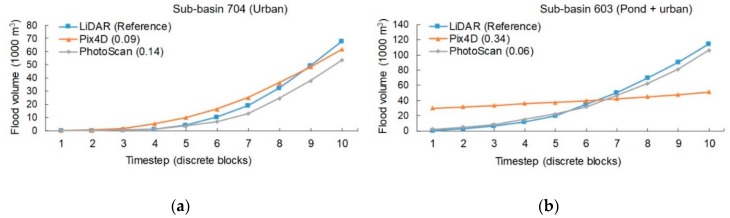
This is a figure, Similarity of flood progression curves compared to LiDAR at 704 (**a**) and 603 (**b**) sub-basins. Bray-Curtis index is shown in brackets. Volume is given in 1k cubic meters (m^3^).

**Table 1 sensors-19-03205-t001:** Summary of technical parameters for UAV-based imaging surveys on the study area.

Parameter	Result
Imagery acquisition date:	17/02/2016 (6 to 10 am); cloudy day and low wind velocity [46].
Flight plan area	5.37 km^2^ (537 ha). Flight 1: 0.9 km^2^; flight 2: 2.2 km^2^ and flight 3: 2.2 km^2^ (see Figure 1):
Ground sample distance, GSD	~ 10.3 cm/ pixel (single image footprint on the ground ~473 m × 355 m)
Flight height:	~ 325 m AGL (height above ground level). Reported by on-board GPS flight log
Overlap/ grid pattern/ strips	80% (longitudinal)/ simple grid [24]/ 19 overlapping strips captured at nadir angle
Number of flights/images:	3 flights, one of 8 min and two of 20 min each, approximately 467 images acquired

**Table 2 sensors-19-03205-t002:** Settings for the SfM-MVS processing chain of UAV-based image with the applied LCPs.

PhotoScan (semi-automatic)		Pix4Dmapper (automatic)	
Parameter	Selected/ Value ^1,2,3^	Parameter	Selected/ Value ^1,2,3^
**Align photos**		**Initial processing**	
Accuracy	“High”	Keypoint image scale	“Full”
Pair selection	“Reference”	Matching image pairs	“Aerial Grid or Corridor”
Reprojection error (pix) ^1^	1.97	Reprojection error (pix) ^1^	0.19
Control point accuracy (pix) ^2^	0.16	Control point accuracy (pix) ^2^	0.65
Camera optimization (%) ^3^	1.90	Camera optimization (%) ^3^	0.21
Sparse cloud (point/m^2^)	0.04	Sparse cloud (point/m^2^)	0.12
**Build dense Cloud**		**Point cloud densification**	
Quality	“Medium”	Point density	“Optimal”
Deep filtering	“Mild”	Min. number of matched	“3”
Dense cloud (point/m^2^)	6.4	Dense cloud (point/m^2^)	6.6
**Classifying dense cloud points**	“Ground points”	**Point cloud classification**	“Classify Point Cloud”
Cell size (m)	“40”	-	-
Max distance (m)	“0.3”	-	-
Max angle (deg)	“5”	-	-
Ground cloud (points/m^2^)	2.09	Ground cloud (points/m^2^)	3.51
**Build mesh**		-	-
Source type	“Height field (2.5D)”	-	-
Point classes		-	-
[Surface mesh]	“Created (Never classified)”	-	-
[Terrain mesh]	“Ground”	-	-
**Build DEM ← [from surface mesh]**		**Raster DSM**	
Source data	“mesh”	**Method**	“Inv. Dist. Weighting”
Interpolation	“Enable (default)”	DSM filters	[all checked]
Resolution	[default value]	Resolution	“Automatic” [1 × GSD]
	40.84 × 40.84 cm		10.35 × 10.35 cm
**Build DEM ← [from terrain mesh]**		**Additional outputs**	
Source data	“mesh”	Raster DTM ^4^	[checked]
Interpolation	“Enable (default)”		
Resolution	[default value]	Resolution	“Automatic” [5 × GSD]
	40.83 × 40.83 cm		51.73 × 51.73 cm

^1^ Quality indicator used as a basis for 3D point reconstruction during bundle block adjustment; values ≤ 1 pix are better. ^2^ Quality indicator (pix) of control point manually tagged on imagery (mean value for 13 LCPs); pix values ≤ 1 are better (the error is less than the average GSD. ^3^ Relative difference (%) between initial and optimized internal camera parameter values (focal distance, pixel), lower values are better. ^4^ DTM (digital terrain model).

**Table 3 sensors-19-03205-t003:** *H* (m) input used for a flood event estimation for each DEM sub-basin.

Sub-Basin	Comment ^1^	LiDAR *H*	PhotoScan *H*	Pix4D *H*
704	Constituted entirely by urban cover (0.32 km^2^); *h =* 1.22	1.48	1.35	1.80
705	Also constituted by urban cover (0.19 km^2^); *h =* 0.29	1.32	0.79	1.28
703	Adjoining outlet of Sub-basin 603 (0.02 km^2^); *h* = 1.78	1.87	1.74	2.02
603	Pond, wetland and urban cover (0.29 km^2^); *h* = 1.63	1.85	2.01	1.63
506	Adjoining inlet of Sub-basin 603 (0.1 km^2^); *h* = 1.12	3.15	3.10	3.78

^1^ Details of simulation flood depth h (m) and sub-basin delineation and its code numbers are explained in Nardini and Miguez [44].

**Table 4 sensors-19-03205-t004:** Accuracy measures of LiDAR based and UAV-derived models (checkpoints = 104).

Accuracy Estimators by Assumption of the *ΔZ* Distribution:	DSM			DEM		
	LiDAR	PhotoScan	Pix4D	LiDAR	PhotoScan	Pix4D
**Normal**						
Mean (m)	0.010	0.165	−0.080	−0.089	−0.073	−0.093
Standard deviation, SD (m)	0.373	0.433	0.284	0.166	0.219	0.262
RMSE ^1^ (m)	0.371	0.462	0.294	0.187	0.230	0.277
RMSE:GSD ^2^ ratio	-	**4.5**	**2.9**	-	**2.2**	**2.7**
**Non-normal (robust method)**						
Median (m)	0.125	0.172	0.142	0.123	0.131	0.135
NMAD ^3^ (m)	0.185	0.255	0.211	0.181	0.185	0.200
NMAD:GSD ^2^ ratio	-	**2.5**	**2.0**	-	**1.8**	**1.9**

^1^ Root mean square error. ^2^ GSD (ground sample distance: 10.3 cm/ pix). ^3^ Normalized median absolute deviation.

**Table 5 sensors-19-03205-t005:** Comparisons of the estimations for a local flood event considered by each DEM.

Sub-Basin	Volume, Area *V* (m^3^), *A* (m^2^)	LiDAR	Pix4D	PhotoScan	Pix4D	PhotoScan
					Difference (%) ^2^	
704 (Urban)	*V*	68,072	61,728	53,699	9.3	21.1
	*A*	166,385	114,940	141,985	30.9	14.7
705 (Urban)	*V*	4,065	185	126	95.4	96.9
	*A*	16,682	941	758	94.4	95.5
703 (Outlet of 603)	*V*	13,986	4,294	7,638	69.3	45.4
	*A*	15,646	5,756	11,793	63.2	24.6
603 (Pond + urban)	*V*	114,311	51,103	106,135	55.3	7.2
	*A*	155,245	21,015	146,596	86.5	5.6
506 (Inlet of 603)	*V*	2,589	5,690	7,180	119.8	177.3
	*A*	7,878	11,206	14,514	42.2	84.2
					**Difference (absolute)**	
Total (Σ) ^1^	*V*	203,023	123,000	174,778	80,023	28,245
	*A*	361,836	153,858	315,646	207,978	46,190
**Overall** **difference (%) ^2^**	***V***				**42.5**	**18.4**
	***A***				**59.3**	**16.4**

^1^ Σ = Total *V* and *A*, corresponding to the sum of the flooded sub-basins. ^2^ Computed with Ecs. 2 and 3.

**Table 6 sensors-19-03205-t006:** Bray-Curtis similarity index of DEM flood estimations with respect to LiDAR reference.

Sub-Basin	Pix4D	PhotoScan	Pix4D	PhotoScan
	Volume ^1^		Area ^1^	
704 (urban)	0.09	0.14	0.16	0.12
705 (urban)	0.85	0.91	0.79	0.85
703 (outlet of 603)	0.47	0.27	0.51	0.19
603 (pond + urban)	0.34	0.06	0.69	0.07
506 (inlet of 603)	0.46	0.54	0.33	0.44
**Overall average score**	**0.44**	**0.38**	**0.50**	**0.33**

^1^ Perfect similarity score is 0, whereas 1 is absolutely dissimilarity.

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
