# Peer review of "DEM Generation from Fixed-Wing UAV Imaging and LiDAR-Derived Ground Control Points for Flood Estimations"

_sensors, 2019, doi:10.3390/s19143205_

Round 1

Reviewer 1 Report

The article solves economically and socially important topic of floods in an urban space that is worth to study by innovative and easy to use and cheap technologies. It is necessary to test current technologies in different environments and for various purposes and therefore this article brings a novelty and is suitable for the journal. Overall quality of results is high. Weakness is its complexity and it seems to be the main reason why the article is not easy to read. Methodology is robust and results cover extensive branch of possible UAV digital models applications in flood survey. Uncommon is that authors solved 3 different aims (2 defined in the article, but according to results it seems they are  three): UAV geotagging using LiDAR – that correspond with the main title of the article “DEM Generation from Fixed-wing UAV Imaging and 2 LiDAR-derived Ground Control Points for Flood Estimations”, further comparison of digital models accuracies in 2 different software applications and the third - authors compared data on flood simulation model what could correspond with the end of the article's title. According to the title of the article it seems that testing of 2 software applications was work done beyond the main scope of the article. Due to complex structure of the article, the discussion and conclusions does not covers sufficiently all aspects following from results and here is a potential for some improvements – notes are in pdf file. However, it is challenging to discussed all 3 aims deeply and analyse possible benefits of valuable results that the article brought to the current problematic.

Some comments shall be implemented to solve unclear terminology or to prove testimony of results. All detailed revision notes are included in pdf file: sensors-538690-peer-review-v1-rev.pdf

-        1. Introduction : define terms DTM and DEM because both were used in the article without any explain why both because in some countries were considered to be identical and in some countries the meaning is different. Further, in the methodology you introduced DSM and a calculation was done from DSM as well – so you need to explain the difference in all 3 terms.

-        Line 44-45 : Define term “flood applications”

-        The main advantage of UAV besides other mentioned issues is also very detailed DEM that can be  acquired from lower height in comparison to laser scanning that usually covers larger areas and does not have a such resolution as data from UAV – this is essential in the assessment of flood risk  - you cited Leitão et al. in the discussion maybe would be worthy and I recommend  to cite his or other relevant work at the beginning of the article, in the introduction because in UAV scans you can see details as curbs on the streets as Leitão et al.  explained: “While it is evident that the representation of roads is critical, requiring a minimum resolution of 2 to 3 m, walls and street curbs are also elements that influence the propagation of a flood wave (Sampson et al., 2012), but to represent these elements in the DEM, a finer resolution ( <1 m) is required. Realistic and detailed representation of terrain thus plays a fundamental role in overland flow modelling.”  Urban flooding conditions are different from these in ambient landscape and therefore a such UAV advantage is visible more in urban areas than under forest canopy for instance.

-        One aim is missing - the comparison of two software applications used for DEM processing - because there is a part in results about this issue - you shall define it in aims of the article and explain why you used both applications not only one software.

-        Lines 153-156 - from the point of view on micro-topography creating drainage channels in urban space you shall define expected accuracy that shall be obtained by modern LiDAR - add a citation, defined quality level less than 20 cm can be OK but there exist works dealing with exact numbers in centimetres what is crucial for flood distribution particullarly in urbanized area - e.g. Zhang et al. "Comparison of TanDEM-X DEM with LiDAR Data for Accuracy Assessment in a Coastal Urban Area"

-        I recommend to rewrite paragraph 4.3 (393-406) “security limitations” can be briefly mentioned but are not necessary to discuss broadly because the article solved different three aims. More important shall be to write a separate paragraph about processing of DTM/DEM by resampling, down-sampling methods - finding optimal sampling distances for the compared digital models. Because it seems that this article that solved only processing of raw data from point clouds but for further flood analysis in GRAS GIS, SAGA GIS ...  it is useful to process primary data from point clouds and prepare adequate DTM/DEM for the comparison of DTMs/DEMs acquired with different technologies. Further, advantages and disadvantages of UAV usage shall be discussed - economic, time point of view of post-processing – for instance which size of an investigated area is economically rentable? Dron shall be applied mainly in areas with potential hazards or existing hazards and a such dataset of the presented case study can be applied in an early warning system and for possible flood volume detection and distribution of water in micro-morphology of streets.

-        In the discussion or in conclusions,  a comparison of both software (Agisoft and Pix4D) is missing. Here shall be stated which one authors recommend to use and why or if the authors think that both applications are worth to use then to explain why - there is a paragraph that could be move here from the results - see in pdf.

T

echnical notes:

Figure 3a- a scheme is repeating from previous figure, remove

Line 214 – Figure1 has wrong number it is Figure 4 - correct

Paragraph in lines 279-284 is doubled.

Figure 7 – numbers are cut in a graph – exchange figure

Figure 8 – enlarge picture – they are too small and content is hardly visible and remove numbers because numbers are in Table 5

Author Response

Response to Reviewer 1 Comments

Review

-       “1. Introduction: define terms DTM and DEM because both were used in the article without any explain why both because in some countries were considered to be identical and in some countries the meaning is different. Further, in the methodology you introduced DSM and a calculation was done from DSM as well – so you need to explain the difference in all 3 terms.”

Response 1: Four lines have been written at the beginning of the introduction to distinguish the difference between the three terms DEM, DSM and DTM.

Review

-        Line 44-45: Define term “flood applications”

Response 2: Response 2: As suggested by the reviewer, the term "flood applications" is clarified   in the context of this work to refer to the estimation of the flooded area and volume.

Review

-        The main advantage of UAV besides other mentioned issues is also very detailed DEM that can be  acquired from lower height in comparison to laser scanning that usually covers larger areas and does not have a such resolution as data from UAV – this is essential in the assessment of flood risk  - you cited Leitão et al. in the discussion maybe would be worthy and I recommend  to cite his or other relevant work at the beginning of the article, in the introduction because in UAV scans you can see details as curbs on the streets as Leitão et al.  explained: “While it is evident that the representation of roads is critical, requiring a minimum resolution of 2 to 3 m, walls and street curbs are also elements that influence the propagation of a flood wave (Sampson et al., 2012), but to represent these elements in the DEM, a finer resolution ( <1 m) is required. Realistic and detailed representation of terrain thus plays a fundamental role in overland flow modelling.”  Urban flooding conditions are different from these in ambient landscape and therefore a such UAV advantage is visible more in urban areas than under forest canopy for instance.

Response 3: According to the reviewer, the cite is relevant and expand the description of the state of the art.

Review

-        One aim is missing - the comparison of two software applications used for DEM processing - because there is a part in results about this issue - you shall define it in aims of the article and explain why you used both applications not only one software.

Response 4: We agree with the reviewer, the suggested goal has been declared, and added to the other two.

Review

-        Lines 153-156 - from the point of view on micro-topography creating drainage channels in urban space you shall define expected accuracy that shall be obtained by modern LiDAR - add a citation, defined quality level less than 20 cm can be OK but there exist works dealing with exact numbers in centimetres what is crucial for flood distribution particullarly in urbanized area - e.g. Zhang et al. "Comparison of TanDEM-X DEM with LiDAR Data for Accuracy Assessment in a Coastal Urban Area"expected accuracy to be obtained by using modern LiDAR (10-cm RMSE)

Response 5: The expected accuracy of the LiDAR was defined based on the suggested reference (Zhang et al. 2019).

Review

-        I recommend to rewrite paragraph 4.3 (393-406) “security limitations” can be briefly mentioned but are not necessary to discuss broadly because the article solved different three aims. More important shall be to write a separate paragraph about processing of DTM/DEM by resampling, down-sampling methods - finding optimal sampling distances for the compared digital models. Because it seems that this article that solved only processing of raw data from point clouds but for further flood analysis in GRAS GIS, SAGA GIS ...  it is useful to process primary data from point clouds and prepare adequate DTM/DEM for the comparison of DTMs/DEMs acquired with different technologies. Further, advantages and disadvantages of UAV usage shall be discussed - economic, time point of view of post-processing – for instance which size of an investigated area is economically rentable? Dron shall be applied mainly in areas with potential hazards or existing hazards and a such dataset of the presented case study can be applied in an early warning system and for possible flood volume detection and distribution of water in micro-morphology of streets.

Response 6: Reviewer's suggestion was accepted. The "security limitations" paragraph was modified and reduced; aspects suggested by the reviewer were added to the discussion (DTM resolution, advantages/disadvantages, potential, etc.).

Review

-        In the discussion or in conclusions, a comparison of both software (Agisoft and Pix4D) is missing. Here shall be stated which one authors recommend to use and why or if the authors think that both applications are worth to use then to explain why - there is a paragraph that could be move here from the results - see in pdf.

Response 7: The comparison between both software was added to the discussion and conclusions. The suggestion of moving a paragraph from the results to the discussion was executed.

Technical notes:

Review

Figure 3a- a scheme is repeating from previous figure, remove

Response 8: The figure was removed

Review

Line 214 – Figure1 has wrong number it is Figure 4 – correct

Response 9: the number of the figure was corrected.

Review

Paragraph in lines 279-284 is doubled.

Response 10: Paragraph in lines 279-284 was deleted

Review

Figure 7 – numbers are cut in a graph – exchange figure

Response 11: The figure was updated.  

Review

Figure 8 – enlarge picture – they are too small and content is hardly visible and remove numbers because numbers are in Table 5

Response 12: The figure was updated.  

The additional comments made in the attached PDF document ("peer-review-4531429.v1.pdf") have been considered entirety. All the corrections made have been indicated using the "Track Changes" function in Microsoft Word. However, some of the modifications are detailed as follows:

Review

L48 “sampling?”

Important

Response 13:  according to “Granshaw, S.I. Photogrammetric Terminology: Third Edition. Photogramm. Rec. 2016, 31, 210–252”,  the term "sample" is correct in this case.

Review

L50-51 sorry I did not understand this sentence, it means that the vertical accuracy less "accurate" than horizontal?”

Response 14: the reviewer has correctly understood the sentence

Review

L78-82 “remove these sentences, it is not a method”

Response 15: We agree with the reviewer (it is not a method), however, this paragraph was kept it because was a text suggestion of the reviewer number 2.

Reviewer 2 Report

Overall timely and interesting research. Some opinions for improvement:

- The English is understandable but includes not a few errors. For example, in the first sentence of the abstract, "DEM" should be "DEMs". The entire draft needs to be checked and corrected.

- The choice of Riohacha City for this research must be explained with a scientific reason, i.e., why the place is suitable for your research topic compared to other places.

- Figure 1 seems to have three map components, and two of them are located within the same box with the same coordinate tics. This is impossible and should be corrected.

- Figure 2 looks a flowchart but three arrows from Stage-1 look strange. Especially one arrow is from an independent box and connected to Stage-3, not Stage-2. Flows in a flowchart should be more simple and straightforward.

- Caption of Fig. 3 needs a general title of the whole figure before "(a)".

- Sections 2.1.1, 2.1.2, and 2.1.4 contain further subsections without numbers such as "first" and "second", corresponding to 2.1.1.1 etc. Such high fragmentation is not good.

- Tables 5 and 6 show sub-basin numbers and they are shown in Fig. 1 but the caption of Fig. 1 does not explain these numbers, which is not good.

- Section 4.3 is unusual because the contents are not about your research this time. You still can write something about regulation, but I recommend not to make a single subsection for it. Perhaps Section 4 can be written without using subsections. 

Author Response

Response to Reviewer 2 Comments

Review

Overall timely and interesting research. Some opinions for improvement:

- The English is understandable but includes not a few errors. For example, in the first sentence of the abstract, "DEM" should be "DEMs". The entire draft needs to be checked and corrected.

Response 1. We agree with the reviewer. Improvements have been made in drafting that have been deemed necessary.

- The choice of Riohacha City for this research must be explained with a scientific reason, i.e., why the place is suitable for your research topic compared to other places.

Response 2 Section 2.1 (description of case study) argues the scientific criteria considered in the selection of the study area.

- Figure 1 seems to have three map components, and two of them are located within the same box with the same coordinate tics. This is impossible and should be corrected.

Response 3  Figure 1 has been corrected

- Figure 2 looks a flowchart but three arrows from Stage-1 look strange. Especially one arrow is from an independent box and connected to Stage-3, not Stage-2. Flows in a flowchart should be more simple and straightforward.

Response 3 According to the reviewer, the suggested changes were made

- Caption of Fig. 3 needs a general title of the whole figure before "(a)".

Response 4 The general title of figure 3 was included.

- Sections 2.1.1, 2.1.2, and 2.1.4 contain further subsections without numbers such as "first" and "second", corresponding to 2.1.1.1 etc. Such high fragmentation is not good.

Response 5 This type of fragmentation was corrected.

- Tables 5 and 6 show sub-basin numbers and they are shown in Fig. 1 but the caption of Fig. 1 does not explain these numbers, which is not good.

Response 6 The explanation of the sub-basin numbers was added to the caption of Figure 1

- Section 4.3 is unusual because the contents are not about your research this time. You still can write something about regulation, but I recommend not to make a single subsection for it. Perhaps Section 4 can be written without using subsections. 

Response 7 In agreement, this section has been rewritten with more relevant aspects for the purposes of this article. The headings subsections have been removed in the discussion section.

This manuscript is a resubmission of an earlier submission. The following is a list of the peer review reports and author responses from that submission.

Round 1

Reviewer 1 Report

The manuscript entitled “ Generation of a medium vertical accuracy DEM from fixed-wing micro-UAV imagery for flood mapping purposes. A case study in data-scarce regions” presented derived DEM from UAV imagery data for flood mapping purpose.

A brief description of the sections before the materials and methods section is needed.

The paper needs a native proofread due to low writing quality.

Writing and spelling corrections are needed. For example:

Line 16: lack   -> the lack

Line 17: Photogrammetry based -> Photogrammetry is based

Line 34  and 35: increasingly is repeated

Line 67: ground sampled distance (GSD) is correct

Line 106: And ?

Line 138: focused instead of focus

Line 141: “references” should be deleted.

Line 242: an important source -> important sources  

Line 302: what is r.lake.xy?

The authors should provide more details for the photogrammetric process in both software.

In addition, the authors should highlight the originality aspects of the paper. 

Reviewer 2 Report

1.       The paper is way too long – it should be shortened and rewritten: there is not much sense in comparing two different software packages unless you test their various settings and/or different scenarios for data collection (e.g. different image overlap, different flying altitudes etc.)

2.       The purpose of the paper must be clearly stated. Is it any bigger research question behind this work? Or it is only a data-driven, application case study? In other words – is it just a local case study, or it will be important and interesting for the international reader (and why)?

3.       Some additional references to studies dealing with the use of UAV in flood monitoring/management are necessary (some examples are provided below)

4.      In most countries, legal limits are 120-150 m a.g.l., which impacts the efficiency of surveys – please, discuss somewhere legal limitation of utilisation of UAVs for flood mapping – it is only briefly mentioned in the discussion. Also, consider VLOS and BVLOS legal requirements.

5.      It would be better to state why did you choose the study area? What is unique about it? Was it only due to the availability of data? Or the area is of some specific interest?

6.      In Figure 2 (Outline of UAV….) you are almost entirely missing organisational and legal aspects: how high are you allowed to fly? How far from the operator? Do you need any permission to form local authorities/airspace control? Etc.

7.      Some additional information about the preparation to the surveys would be necessary –

Did you have permissions required to fly at 325 m altitudes? Did you have permission required to operate beyond visual line of sight? Did you contact local authorities before the survey? Did you contact airspace controller? Etc.

8.      L. 163: It is not possible to generate valid DEM with GSD equal to the GSD of orthomosaic (in your case 0.1 m) – there is just not enough information in the images. Please, address this issue.

9.      L.214-230 and section 2.4  –control points derived from LIDAR data were used to process UAV data. Then, the same LIDAR dataset where compared with photogrammetry products to asses accuracy – such approach does not make much sense, as you should compare UAV-derived products with some independent dataset, not the one from which control points were generated. Not to mentions also a difference in resolution (1 m LIDAR vs 0.1 m UAV-data).

10.   “beyond the scope of this study” – this sentence appears several times. One may wonder, what actually was the scope of your study? It would be better to avoid such sentences.

11.   Figure 7 – all distributions clearly show that there is a systematic error in your data, i.e. all UAV-generated DEMs and DTMs are “higher” than LIDAR data – it is not a normal distribution. It is also confirmed by the results of your t-test.

Moreover, paper in its current form is sometimes hard to follow due to imprecise language. I have added some detailed comments to the first chapter, but there is much more of them. I would suggest making the paper much shorter and precise case study and focus on its applied aspect.

Examples of studies were UAV employed to study floods:

Şerban, G., Rus, I., Vele, D., Breţcan, P., Alexe, M. and Petrea, D., 2016. Flood-prone area delimitation using UAV technology, in the areas hard-to-reach for classic aircrafts: case study in the north-east of Apuseni Mountains, Transylvania. Natural Hazards, 82(3), pp.1817-1832.

Coveney, S. and Roberts, K., 2017. Lightweight UAV digital elevation models and orthoimagery for environmental applications: data accuracy evaluation and potential for river flood risk modelling. International journal of remote sensing, 38(8-10), pp.3159-3180.

Watanabe, Y. and Kawahara, Y., 2016. UAV photogrammetry for monitoring changes in river topography and vegetation. Procedia Engineering, 154, pp.317-325.

Langhammer, J., Lendzioch, T., Miřijovský, J. and Hartvich, F., 2017. UAV-based optical granulometry as tool for detecting changes in structure of flood depositions. Remote Sensing, 9(3), p.240.

Langhammer, J., Bernsteinová, J. and Miřijovský, J., 2017. Building a high-precision 2D hydrodynamic flood model using UAV photogrammetry and sensor network monitoring. Water, 9(11), p.861.

Hervouet, A., Dunford, R., Piégay, H., Belletti, B. and Trémélo, M.L., 2011. Analysis of post-flood recruitment patterns in braided-channel rivers at multiple scales based on an image series collected by unmanned aerial vehicles, ultra-light aerial vehicles, and satellites. GIScience & Remote Sensing, 48(1), pp.50-73.

Tamminga, A.D., Eaton, B.C. and Hugenholtz, C.H., 2015. UAS‐based remote sensing of fluvial change following an extreme flood event. Earth Surface Processes and Landforms, 40(11), pp.1464-1476.

Perks, M.T., Russell, A.J. and Large, A.R., 2016. Advances in flash flood monitoring using unmanned aerial vehicles (UAVs). Hydrology and Earth System Sciences, 20(10), pp.4005-4015.

Flener, C., Vaaja, M., Jaakkola, A., Krooks, A., Kaartinen, H., Kukko, A., Kasvi, E., Hyyppä, H., Hyyppä, J. and Alho, P., 2013. Seamless mapping of river channels at high resolution using mobile LiDAR and UAV-photography. Remote Sensing, 5(12), pp.6382-6407.

Witek, M., Jeziorska, J. and Niedzielski, T., 2014. An experimental approach to verifying prognoses of floods using an unmanned aerial vehicle. Meteorology Hydrology and Water Management. Research and Operational Applications, 2.

Niedzielski, T., Witek, M. and Spallek, W., 2016. Observing river stages using unmanned aerial vehicles. Hydrology and Earth System Sciences, 20(8), pp.3193-3205.

Langhammer, J. and Vacková, T., 2018. Detection and mapping of the geomorphic effects of flooding using UAV photogrammetry. Pure and Applied Geophysics, 175, pp.3223-3245.
